# The Impact of Dance Movement Interventions on Psychological Health in Older Adults without Dementia: A Systematic Review and Meta-Analysis

**DOI:** 10.3390/brainsci13070981

**Published:** 2023-06-22

**Authors:** Odile Sophie Podolski, Tim Whitfield, Leah Schaaf, Clara Cornaro, Theresa Köbe, Sabine Koch, Miranka Wirth

**Affiliations:** 1German Center for Neurodegenerative Diseases (DZNE), 01307 Dresden, Germanymiranka.wirth@dzne.de (M.W.); 2Division of Psychiatry, University College London, London W1T 7NF, UK; tim.whitfield@ucl.ac.uk; 3Research Institute for Creative Arts Therapies (RIArT), Alanus University of Arts and Social Sciences, 53347 Alfter, Germany; 4Faculty of Therapy Sciences, SRH University Heidelberg, 69123 Heidelberg, Germany; 5Faculty of Fine Arts and Music, CAMTRU, University of Melbourne, Melbourne, VIC 3006, Australia

**Keywords:** non-pharmacological intervention, prevention, randomized clinical trials, dance, aging, mild cognitive impairment

## Abstract

Background: Lifestyle-based multimodal interventions that integrate physical, sensory, cognitive and social enrichment are suggested to promote healthy mental aging and resilience against aging and Alzheimer’s disease (AD). Objectives: This meta-analysis examined the efficacy of dance movement interventions (DMI) as an integrated mind–body activity on outcomes of psychological health in older adults. Methods: Pre-registration was carried out with PROSPERO (CRD42021265112). PubMed, Web of Science and PsycINFO were searched for randomized controlled trials (RCT) evaluating the effects of DMI (>4 weeks’ duration) compared to comparators on measures of psychological health (primary outcome) and cognitive function (additional outcome) among older adults without dementia (aged ≥55). Data of 14 primary RCT (*n* = 983, *n-DMI* = 494, *n-control* = 489) were synthesized using a random effects meta-analysis with robust variance estimation. Results: DMI had a small positive effect on overall psychological health (*g* = 0.30; 95% confidence interval [*CI*]: 0.06, 0.53; *p* = 0.02, *I*^2^
*=* 65.04) compared to control conditions. Small effects of DMI on positive and negative psychological domains as well as quality of life were not statistically significant. DMI had a medium positive effect on general cognitive function (*g* = 0.50; 95% *CI*: 0.12, 0.89, *p* = 0.02, *I*^2^
*=* 79.61) over comparators. None of the primary intervention studies evaluated measures of neuroplasticity. Conclusions: We found that DMI was effective in promoting mental health amongst older adults without dementia, suggesting that the multimodal enrichment tool is a potential strategy for health promotion and prevention of AD. High-quality intervention studies are needed to expand evidence on DMI-induced changes in specific psychological domains and identify underlying neurophysiological correlates.

## 1. Introduction

Due to longer life expectancy, the global population of older adults continues to grow [1]. As a result, the prevalence of age-related conditions, such as Alzheimer’s disease (AD), has increased. Whilst disease-modifying therapies are emerging, they are not curative, and thus the increased prevalence of AD (the most common cause of dementia) poses tremendous economic and social challenges to healthcare systems, patients and their caregivers [2,3,4]. Lifestyle-related physical, cognitive and psychosocial modifiable risk factors contribute to aging processes and the development of age-related conditions [5,6]. Psychological risk factors, including depression, anxiety, stress and social isolation, have been proposed to increase the risk of cognitive impairment and dementia [5,7], possibly via depleting resilience against brain pathology and accelerating cognitive decline [7]. As a consequence, there is an urgent need to establish effective behavioral strategies that target these health risk factors and promote healthy aging, which may ultimately contribute to the prevention of AD [8].

Mental or psychological health can be considered a key target for healthy aging and AD prevention strategies [5]. Therefore, it is emphasized that mental health is a psychological state of well-being and psychosocial integration—not just the absence of mental disease [9]. This includes fostering personal coping strategies as well as purpose in life and personal growth as individual resources [10]. Indeed, studies in older adults have indicated that negative psycho-affective burden is related to elevated brain pathology [11], altered brain functioning [12,13], greater cognitive decline and increased risk of developing AD [11,14,15]. Age-related health conditions are also closely related to psychological health and can lead to increased feelings of negative affect and, as a result, lower perception of quality of life or well-being [16,17]. In contrast, psychological well-being has been associated with a lower prevalence of age-related health concerns, including cardiovascular disease, cognitive failure or physical dysfunction [18,19,20,21]. Meta-analyses have further provided evidence that supports an association between higher well-being and reduced mortality risk [22,23]. Moreover, in older populations, positive psychosocial factors, including social engagement, mindfulness, stress resilience and positive thinking styles, such as optimism or the sense of having a purpose in life, have been associated with better structural and functional brain integrity as well as cognitive health [24,25,26], reduced risk of cognitive impairment and dementia [18,27,28].

In general, it can be assumed that mental health positively influences healthy and resilient aging on several levels, including physical and psychological levels [29]. A theoretical framework is given in a comprehensive article on resilient aging [29], conceived by using the previous work by Kubzansky and colleagues [30]. Based on this biopsychosocial model, three main pathways, in which psychological and social well-being influence health status and act as resilience factors against accumulated stressors associated with aging, are described. These three pathways comprise the promotion of (1) psychosocial factors that buffer adversities of stress or harmful events, (2) health behaviors and (3) positive biological processes. It is proposed that improving well-being in older adults works through these multiple pathways to reduce vulnerability to age-related conditions.

To effectively promote mental or psychological health, it is important that lifestyle-based interventions target mental/psychological, physical, and social well-being. Promoting these three key dimensions of a person’s overall health status [9] is likely to require multimodal or multi-domain interventions, as they combine multiple beneficial lifestyle activities [8]. In this line, studies in animal models have demonstrated far-reaching benefits of environmental enrichment; this typically integrates multimodal stimulation of motor, sensory, cognitive and social processes [31]. Lifestyle activities suggested to resemble multimodal enrichment in humans integrate physical (body) and mental (mind) activities to encourage an “*embodied mind in motion*” [32]. A prime example is dance or dance movement interventions (DMI), which combine music (sensory) with movement (physical) and mental stimulations in a socially engaging environment. Amongst others, DMI may comprise diverse programs, including traditional dance, aerobic dance and dance/movement therapy. The latter is part of the creative arts therapies and incorporates psychological factors and techniques, such as embodiment, creativity, emotional processing/reflection and psychosocial integration, that are integral to mental health and well-being [33]. DMI may thus offer an all-in-one enrichment tool that could promote healthy aging of physical, psychological and social functioning in older populations [34,35].

Given the importance of psychological risk factors in the development and progression of cognitive impairment and dementia [5], advanced knowledge of the impact of DMI on mental or psychological health is needed. Existing evidence has suggested that engaging in regular dance activities is associated with a decreased risk of dementia [36]. Evidence syntheses have further indicated robust positive benefits of DMI versus comparators on cognitive health [37,38,39] and physical/physiological health [40,41,42] in older populations. Yet, evidence on the impact of DMI on psychological health and well-being, specifically in older adults without dementia, remains sparse and inconclusive. According to existing meta-analyses in a broad sample of studies, including younger, middle-aged and older populations, DMI improves overall psychological health as well as positive and negative psychological functions [43,44]. Other meta-analyses have shown that DMI may reduce depression [45] and improve quality of life (QoL) [46], specifically in older non-demented adults, although a different review observed mixed evidence of efficacy across psycho-behavioral outcomes [47]. 

The main objective of the present study was to conduct a systematic review and meta-analysis of the efficacy of DMI versus comparators for improving psychological health outcomes in older adults without dementia (i.e., clinically normal, subjective cognitive decline (SCD) or mild cognitive impairment (MCI)) using data from randomized controlled trials (RCT). Furthering knowledge on this central health aspect may inspire the design of new-generation DMI, aiming to improve mental health and well-being in older populations. 

## 2. Material and Methods

### 2.1. Protocol and Registration

This systematic review and meta-analysis were conducted following the Preferred Reporting Items for Systematic Reviews and Meta-Analyses (PRISMA) recommendations [48] and was registered with PROSPERO in July 2021 (CRD42021265112).

### 2.2. Criteria for Considering Studies for this Review

#### 2.2.1. Types of Studies

RCT were considered eligible for this review. To be included, trials had to clearly describe that participants were allocated via individual or cluster randomization.

#### 2.2.2. Types of Participants

The population of interest for this review was adults aged 55 or older without a diagnosis of dementia. Thus, clinically normal older participants or participants with a diagnosis of SCD or MCI were included. To ensure that clinical samples were ascertained according to standardized protocols, diagnostic criteria had to be reported. No restrictions regarding the type of living arrangement were made. Studies with an explicitly defined target population of patients with current or past diagnoses of psychiatric, neurological, inflammatory and/or other serious medical disorders (e.g., dementia, diabetes, cancer) were excluded.

#### 2.2.3. Types of Interventions

After carefully revising previous meta-analyses and consulting experts in the field, we conceptualized DMI as multimodal mind–body activities that involve coordinated movements carried out in synchronization with music and progressing through space. We considered a broad variation of DMI and dance movement styles eligible, as these activities inherently share common basic principles, including simultaneous stimulation of multimodal motor, sensory, cognitive, social and emotional processes [35,44]. Studies had to include a DMI incorporating a pre-specified training program in any style, for example, traditional/ social dance, creative/expressive/ meditative dance, dance/movement therapy or eurythmy. Individual and group interventions were eligible, including both professionally facilitated and self-guided formats. DMI in a virtual or video game format were also eligible for inclusion, although none were eligible following full-text screening. To be included, interventions had to have a minimum duration of 4 weeks. No restrictions for session frequency were applied. Study comparators could be active or passive. Interventions consisting solely of physical exercises in terms of aerobic strength training, coordination training, ergometer training or meditation and breathing exercises were excluded. 

#### 2.2.4. Types of Outcome Measures

##### Primary Outcome

The primary outcome of the study was psychological health. This outcome was evaluated through self-reported (subjective) and validated assessment tools. Due to the limited data on the effects of DMI on psychological health in the target population, we utilized a broad clustering of psychological outcome measures for the present analysis. This methodological approach was selected in accordance with previous meta-analytical works [43,44]. Firstly, we thus synthesized all outcome measures across DMI studies to assess the impact of DMI versus comparators on overall psychological health. In addition, we created two subcategories of psychological health, thereafter termed psychological domains. The ‘positive’ domain subsumed measures associated with positive mood and emotions, such as psychological well-being and social integration/connectedness. The ‘negative’ domain subsumed measures related to negative mood and emotions, such as depression, anxiety, and psychological distress. Beyond that, measures related to QoL were subsumed, including health-related QoL and lifestyle factors.

##### Additional Outcome

As an additional outcome, we assessed general cognitive function, measured through validated neuropsychological measures of general cognitive ability. Given this focus, only neuropsychological instruments associated with global cognitive status and/or global cognitive abilities were considered, including the Mini-Mental State Examination (MMSE) [49], the Montreal Cognitive Assessment (MoCA) [50] and the Raven’s Progressive Matrices Test [51].

### 2.3. Search Methods for Identification of Studies

The following electronic databases were searched: PubMed, Web of Science and PsycINFO. Eligible studies had to have full-text availability and be written in English and were restricted to peer-reviewed journals and academic publications (dissertations). The initial search was conducted on 13 April 2021 and updated on 10 November 2021 and 23 April 2023, respectively. Key terms and search strings were derived from pre-existing systematic reviews and meta-analyses with related topics [38,42,52,53] and adapted to the present research question. A combination of keywords, free terms and Medical Subject Headings (MesH) terms was applied across databases. Search strings combined words covering the main elements of the research question, including the stems “dance*” or “music and exercise” with “older adults” or “psychological health” and interventional terms such as “random*”. Database-specific search strings are provided in the Appendix A. 

#### 2.3.1. Selection of Studies

After deduplication with the reference manager software EndNote (version X9.3.1), a two-stage screening process was utilized. After a pilot screening, two trained independent reviewers (OP, LS), unblinded to the study authors, screened the identified studies in parallel. At both the title–abstract and full-text stages, studies were evaluated according to the specified inclusion and exclusion criteria. To ensure high standardization and facilitate screening, a standardized operating procedure (SOP) for the reviewing process was created. Each stage of the screening was completed by comparing decisions of the reviewers. Where no consensus could be reached by discussion, a third independent reviewer (TK, MW) was engaged for the final decision on eligibility. Reasons for exclusion of studies were documented. A PRISMA flow diagram [48] was created to display the study selection process. Cohen’s kappa was calculated to assess inter-rater reliability. 

#### 2.3.2. Data extraction and Management

Two trained independent reviewers (OP, LS) used a standardized form to extract all study data in parallel. The information extracted covered general information (i.e., study authors, publication date and type, recruitment country, study funding and conflicts of interest), methodological information (i.e., study design and setting, characteristics of participants, descriptions of the intervention and comparator) and outcome information (i.e., outcomes of interest, measurement tools, outcome data reported pre- and post-intervention). Discrepancies between the data extraction forms were discussed until a consensus was reached. Where outcome data were reported using the standard error (SE), a conversion to standard deviation (SD) was performed in order to calculate effect sizes. Studies reporting insufficient data for statistical synthesis were excluded. Where appropriate, study authors were contacted for additional information via e-mail.

#### 2.3.3. Assessment of Risk of Bias in Included Studies

For each eligible study, risk of bias was assessed during data extraction with the Risk of Bias 2 tool [54] of the Cochrane Collaboration. Each study was rated according to an intention-to-treat approach and with respect to the primary outcome (i.e., psychological health) only. Each of the following domains (D) was rated in parallel by two independent reviewers (OP, LS) as either “high risk”, “some concerns” or “low risk”: (D1) bias arising from the randomization process; (D2) bias due to deviations from intended interventions; (D3) bias due to missing outcome data; (D4) bias in measurement of the outcome; and (D5) bias in selection of the reported result. The overall bias judgement is reached by an algorithm based on the answers given to several signaling questions within each domain. Disagreement was resolved by discussion until consensus was reached.

#### 2.3.4. Assessment of Heterogeneity

The parameter *I*^2^ was used to assess heterogeneity within each model. The *I*^2^ statistic represents the percentage of variability in effect sizes not attributable to sampling error. Interpretation followed a previously described rule of thumb [55] with *I*^2^ = 25% meaning low heterogeneity, *I*^2^ = 50% moderate heterogeneity and *I*^2^ = 75% substantial heterogeneity.

#### 2.3.5. Assessment of Reporting Bias

A funnel plot created in R using the “metafor” package (version 3.0-2) was used to graph the observed effect sizes against standard errors. Interpretation and assessment of asymmetry were supported by employing Egger’s regression test [56], indicating publication bias through a significant result. 

### 2.4. Data Synthesis

For the calculation of effect sizes for controlled designs with pre- and post-test scores, the standardized mean difference (SMD with 95% confidence intervals (*CI*)) with a bias correction (*g*) was used in accordance with the reference literature [57,58,59]. The outcome assessments with lower scores indicating an improvement after DMI were inverted, such that higher values reflected improvement or better performance for any given outcome. The calculations (see Appendix A for detailed descriptions and formulas) were applied and implemented using the free statistical computation software R (version 4.0.3) for each outcome score to determine effect sizes.

#### 2.4.1. Accounting for Dependencies

As defined in the inclusion criteria, the outcomes of interest could be assessed via multiple assessment tools within a given primary study. Incorporating more than one outcome measure per study violates the assumption of independent effect sizes of common meta-analysis models [60]. To overcome this problem, typically, only one effect size or an average effect size per study is selected. These procedures result in loss of information and should therefore be avoided [61]. To adequately account for dependencies in outcome measures within studies, we used a random-effects meta-analysis with robust variance estimation (RVE) [62]. Modelling was implemented through the “robumeta” package (version 2.0) in R with a correction for small samples [63]. The package provides a default setting of *rho* at 0.8 for the assumed correlation of outcomes within studies. Sensitivity analyses varying *rho* from 0–1 were run to investigate the possible impact on *Tau*^2^ as estimator of variance. 

Results were defined as being significant with a *p*-value of < 0.05. Where degrees of freedom fall below 4, *p*-values for RVE meta-analytic estimates are considered unreliable [63] and are therefore not reported. The following rubric was used to interpret effect sizes: *g* ≥ 0.2 (small), *g* ≥ 0.5 (medium) and *g* ≥ 0.8 (large) [64]. Forest plots were created in R to visualize results.

#### 2.4.2. Further Analyses

As the number of included primary studies was limited, a meta-regression analysis was omitted in the present synthesis. An exploratory post hoc outlier analysis was carried out. For this, effect sizes of outcomes with *CIs* outside the boundaries of the pooled effect *CI* across studies were classified as outliers [65]. Additionally, when in doubt of methodological quality or where high probability of bias occurred, individual studies were excluded from exploring influence on effects. Subsequently, analyses described before were repeated, excluding studies classified as outliers. 

## 3. Results

### 3.1. Study Selection

The initial systematic search across the three databases in April 2021 yielded a total of 640 records. After deduplication, 484 records remained for the title–abstract screening. Two independent reviewers (OP, LS) screened the titles and abstracts with an inter-rater reliability (unweighted Cohen’s kappa) of 0.80 (95% *CI* [0.73, 0.88]). Studies meeting eligibility criteria were full-text screened by OP and LS in parallel, reaching a high agreement with an unweighted kappa of 0.93 (95% *CI* [0.84, 1.02]). During the process of data extraction, 15 studies were excluded. Reasons for exclusion were documented. Study selection resulted in a final number of 12 primary studies (see Figure 1 for PRISMA flowchart). An additional primary study was included after the search update in November 2021 and April 2023, respectively, resulting in a total of 14 studies.

### 3.2. Study Characteristics

An overview of study characteristics is provided in Table 1. Publication years ranged from 2009 to 2022. Seven studies (50%) were conducted in Europe, five studies (36%) in Asia and one study (7%) in Canada and Brazil, respectively. Thirteen peer-reviewed journal articles and one dissertation [66] were included. One study [67] was a secondary analysis of an RCT reported elsewhere [68]. Psychological health measures were a primary outcome in only one study [69] and were otherwise included as secondary outcomes. The majority of studies used cognitive function (*k* = 8; 57%) or physical function (*k* = 5; 36%) as the primary outcome.

Most of the studies employed randomization at the individual level (*k* = 12; 86%); one study used cluster randomization [69]. Another study [70] divided participants into two groups prior to randomization to maintain social peers. Two trials applied a three-armed design with two control group types [66,71]. Four studies (29%) reported several time points of assessment during the intervention [69,71,72,73]. Only one study [74] reported a follow-up assessment beyond the end of the intervention.

**Table 1 brainsci-13-00981-t001:** Characteristics of included studies.

Study	Country	Participants				Intervention				Outcomes
		N (EXP/AC/PC)	Mean Age	Descriptions	Proportion of Females	Type of DMI	Period and Frequencies	Intensity	Control Type	
Alves, 2013 [66]	Brazil	65 (25/15/25)	68.1	non-clinical	92.3%	Ballroom Dance	16 weeks, 2/week, 120 min,	-	AC: Walking PC: no contact	(1) Ryff’s PWBS (2) PSQI (3) BAI (4) PSS (5) Raven’s Advanced Matrices
Bisbe et al., 2020 [75]	Spain	31 (17/14/-)	75.1	MCI	48.4%	Choreography	12 weeks, 2/week, 60 min	light–moderate	AC: Physical Therapy	(1) SF-36 (2) HADS-A (3) HADS-D (4) MMSE
Chang et al., 2021 [73]	China	109 (62/47/-)	76.3	MCI/SCD	100%	Square Dance	18 weeks, 3/week, 30 min	low	PC: Usual Care	(1) SF-12 (2) GDS-15 (3) MoCA
Cruz-Ferreira et al., 2015 [76]	Portugal	57 (32/-/25)	72.0	non-clinical	100%	Creative Dance	24 weeks, 2/week, 50 min	-	PC: Waitlist	(1) LSS
Esmail et al., 2020 [71]	Canada	41 (12/15/14)	67.5	non-clinical	75.6%	Dance Movement	12 weeks, 3/week, 60 min	-	AC: Aerobic Exercise PC: Waitlist	(1) SF-12 (2) HPLP2 (3) MHC (4) LSNS (5) STAI-Trait (6) BPI (7) MoCA
Eyigor et al., 2009 [77]	Turkey	37 (19/-/18)	72.4	non-clinical (depression included)	100%	Folkloric Dance	8 weeks, 3/week, 60 min	-	PC: no intervention	(1) SF-36 (2) GDS
Hars et al., 2014 [67]	Switzerland	134 (66/-/68)	75.5	non-clinical (age-related medical conditions included)	96.3%	Eurythmy	25 weeks, 1/week, 60 min	-	PC: Waitlist	(1) HADS-A (2) HADS-D (3) MMSE
Hui et al., 2009 [70]	China	97 (52/- /45)	68.0	non-clinical	96.9%	Aerobic dance	12 weeks, 2/week, 50 min	low	PC: No intervention	(1) SF-36
Kosmat and Vranic, 2017 [74]	Croatia	24 (12/12/-)	80.8	non-clinical	62.5%	Standard Dance	10 weeks, 1/week, 45 min	-	AC: Social discussion	(1) SWLS (2) GSE
Lazarou et al., 2017 [78]	Greece	129 (66/-/63)	66.8	MCI	78.3%	Ballroom Dance	40 weeks, 2/week, 60 min	-	PC: No intervention	(1) NPI (2) MMSE, MoCA
Liao et al., 2018 [69]	China	107 (55/52/-)	71.8	non-clinical (mild–moderate depressive symptoms)	61.7%	Music and Tai-Chi (combined)	12 weeks, 3/week, 50 min	moderate	AC: Routine health education	(1) GDS
Mishra et al., 2022 [79]	India	40 (20/20/-)	65.7	non-clinical	80%	Folkloric Dance	6 weeks, 5/week, 60 min	moderate	AC: Exercise program	(1) SF-36 (2) MoCA
Serrano-Guzmán et al., 2016 [80]	Spain	52 (27/25/-)	69.3	non-clinical	100%	Dance Therapy (Flamenco)	8 weeks, 3/week, 50 min	low impact	AC: Self-care treatment advice	(1) SF-12
Zhu et al., 2018 [72]	China	60 (29/-/31)	69.6	MCI	60%	Aerobic Dance	12 weeks, 3/week, 35 min	moderate	PC: No intervention	(1) GDS-15 (2) SF-36 (3) MoCA

Note. Sample sizes refer to participants analyzed in the primary studies; outcomes listed only refer to outcomes included in quantitative syntheses of data; “-“ indicates that no information could be extracted from the study paper; Key: N, sample size; EXP, intervention group; AC, Active Control; PC, Passive Control; PWBS, Psychological Well-Being Scales; BAI, Beck Anxiety Inventory; PSS, Perceived Stress Scale; PSQI, Pittsburgh Sleep Quality Index; MCI, mild cognitive impairment; HADS-A, Hospital Anxiety and Depression Scale—Anxiety; HADS-D, Hospital Anxiety and Depression Scale—Depression; SF-36, 36-Item Short Form Health Survey; MMSE, Mini-Mental State Examination; SCD, subjective cognitive decline; SF-12, 12-Item Short Form Health Survey; GDS, Geriatric Depression Scale; MoCA, Montreal Cognitive Assessment; LSS, Life Satisfaction Scale; STAI-Trait, State-Trait Anxiety Inventory—Trait; HPLP2, Health Promoting Lifestyle Profile -2; MHC, Mental Health Continuum Short Form; BPI, Brief Pain Inventory; LSNS, Lubben Social Network Scale; SWLS, Satisfaction with Life Scale; GSE, General Self-Efficacy Scale; NPI, Neuropsychiatric Inventory.

### 3.3. Participant Characteristics

This review summarized data from 983 participants (*n-DMI* = 494, *n-control* = 489). Sample sizes varied from 24 to 134 participants per study. Participants enrolled in the primary studies had an average age of 71 years. The average percentage of female participants in the reported samples was 82.3 %. Four studies (29%) included female participants only. Nine studies (64%) recruited healthy older adults; one study included age-related health conditions [67]. The remaining four studies (28%) investigated participants with a confirmed diagnosis of MCI [72,73,75,78]. None of the primary studies explicitly investigated older adults with SCD. Five studies (36%) provided information on education in years with an average of 11 years of education.

Overall, it was reported that included participants neither had the experience of dance nor engaged in similar activities before starting the trial. Of note, 10 studies (71%) reported specific assessments of activity levels or whether participants regularly engaged in physical exercise prior to intervention. Five studies (36%) examined a sample defined as inactive [71] or (moderately) sedentary [66,73,75,80], whereas two studies examined participants with an active lifestyle prior to intervention [69,77]. 

### 3.4. Intervention Characteristics

#### 3.4.1. Intervention Type/Types of DMI

A comprehensive overview of the intervention characteristics is provided in the Appendix A. Types of DMI varied across the studies. Six studies (43%) included standard or traditional dance (e.g., ballroom dance, folkloric dance) [66,73,74,77,78,79]. Three studies (21%) reported aerobic dance [70,72,75]. Five studies (36%) were characterized as evaluating creative arts therapies [33]. Those included interventions designed according to the standards of the American Dance Therapy Association [71,81] and a creative dance program [76]. Hars and colleagues [67] employed eurythmy to apply multi-task practices to piano music and Liao and colleagues [69] used Tai-Chi movements combined with music. 

#### 3.4.2. Intervention Period, Duration and Frequency

The median intervention duration was 12 weeks (range: 6 to 40 weeks). Ten studies (71%) had a length of 16 weeks or below [66,69,70,71,72,74,75,77,79,80]. Four studies (28%) were conducted over 18 weeks or more [67,73,76,78]. Frequencies of DMI sessions varied from once per week to five times per week, while individual session durations lasted from 35 to 120 min. The training dose ranged from a minimum of ten DMI sessions with 45 minutes’ duration [74] to a maximum dose of 80 sessions of 60 min [78]. 

Across studies, the structure of all DMI sessions comprised a warm-up, followed by the training phase and, subsequently, a cool-down or stretching phase. Notably, one study [77] required participants to walk for approximately 30 min twice a week in addition to the DMI. Information regarding the intensity of DMI sessions was presented in seven studies (50%). Three studies (21%) had low intensity [70,73,80], and four had (28%) moderate intensity sessions [69,72,75,79]. One study ensured that the target intensity was achieved by measuring the individual heart rates during training [72].

#### 3.4.3. Intervention Setting

All studies performed DMI in a group setting. In three studies (21%), the DMI was designed to encourage social interaction between pairs or group members [66,76,77]. For the remaining interventions, each person performed the exercises individually. Eight studies (57%) were conducted at local healthcare centers, hospitals or similar public facilities. One DMI took place in an outdoor setting [73]. Most of the interventions (*k* = 11; 79%) were led by a certified or experienced (dance) instructor or a facilitator specially trained for the DMI program. Three studies (21%) [69,79,81] did not provide information on instructor/facilitator qualifications.

#### 3.4.4. Intervention Adherence

Nine studies (64%) provided data on adherence to the DMI, mainly reported as the proportion of classes or courses attended. Zhu and colleagues (60) reported the median number of sessions attended. Across studies, DMI attendance rates ranged from 79% to 100%; five studies (36%) reported adherence of 90% or more [66,70,71,75,81]. Across studies, the proportion of individuals who discontinued study participation ranged from 0 to 48% for intervention and control groups. Two studies reported that all participants completed the trial [76,81]. Three studies (21%) were unclear regarding the proportion of participants who dropped out. 

#### 3.4.5. Control Condition

Twelve studies (86%) included a single comparator condition (active or passive), whereas two studies included both an active and passive control group [66,71]. Of the twelve studies employing one comparator, seven evaluated DMI compared to a passive control group with waitlist (*k* = 2) [67,76], usual care (*k* = 1) [73], or no intervention conditions (*k* = 4) [70,72,77,78]. Five studies utilized an active comparator, including physical therapy [75], exercise program [79], discussion groups [74], routine health education [69] and self-care advice [80].

### 3.5. Neurophysiological Measures

None of the primary studies has evaluated indicators of physiological changes and neuroplasticity.

### 3.6. Risk of Bias

The methodological quality of the primary studies varied. A visual summary of the risk of bias ratings is provided in Figure 2. The overall risk of bias was rated as being at “high risk” in nine studies (64%), while five studies (36%) were rated as having “some concerns”; no study was rated as “low risk”. Please see the Appendix A for domain-specific risk of bias ratings. 

### 3.7. Publication Bias

The distribution of effect sizes and SEs was visualized using a funnel plot (see Appendix A). Whilst a degree of asymmetry was apparent on the funnel plot, an Egger’s test found no evidence of publication bias (intercept, −0.631 *p* = 0.54; *CI* 95% [−0.11, 1.18]).

### 3.8. Quantitative Synthesis of Results

#### 3.8.1. Primary Outcome

A detailed overview of the outcome measures that were included in the present data synthesis is provided in the Appendix A. Across studies, DMI significantly outperformed comparators for overall psychological health with a small effect size (*g* = 0.30, *p* = 0.02; see Table 2 and forest plot in the Appendix A). The result was based on the pooled data of psychological health outcomes (*k* = 14) and compared to passive control conditions (*k* = 9), where applicable (e.g., for studies utilizing both passive and active comparators). The *I*^2^ statistic indicated a moderate amount of heterogeneity among the included studies. Sensitivity analyses varying the parameter *rho* did not change the value of *Tau*^2^, representing robust variance estimation against dependencies. 

The separation of psychological health outcomes into positive and negative domains showed a non-significant trend in favor of DMI over comparators (see Table 2) with a small effect size in well-being (*g* = 0.30, *p* = 0.07). There was a large positive effect of DMI on social integration (*g* = 0.85); however, degrees of freedom fell below four resulting in a low reliability of effect size. The small effects of DMI on anxiety/stress and depression were not statistically significant. In addition, there was a small effect of DMI on QoL compared to controls conditions that was statistically non-significant (see Table 2 and forest plot in the Appendix A). Including all scales in the meta-analytical model revealed a small positive overall effect in favor of DMI over control conditions (*g* = 0.32, *95% CI* [0.12, 0.52]; *p* = 0.005; *I^2^* = 58.10). Both positive and negative domains showed moderate to substantial (61–85%) heterogeneity. Sensitivity analysis varying *rho* did not impact variance estimation.

#### 3.8.2. Additional Outcome: Cognitive Function

As expected, DMI outperformed comparators for general cognitive function, with a medium effect size (*g* = 0.50, *p* = 0.02; see Table 2). The observed heterogeneity was moderate. Excluding the Raven’s Matrices Scale in a post hoc analysis revealed a similar effect size, while heterogeneity remained substantial (*g* = 0.47; 95% *CI* [0.02, 0,92]; *p* = 0.04; *I^2^* = 82.3)

#### 3.8.3. Additional Analyses

Due to the small number of studies, a meta-regression analysis was not feasible. Results of the outlier-adjusted meta-analyses are given in the Appendix A. The exclusion of six individual effect sizes corroborated our results for overall psychological health (*g* = 0.37, *p* < 0.01) as well as for well-being (*g* = 0.46, *p* < 0.01) and QoL (*g* = 0.37, *p* < 0.01) domains. Heterogeneity assessed with the *I*^2^-statistic nominally decreased to a small amount, justifying the exclusion of the respective scales. Post hoc exclusion of one study [70] due to methodological considerations further substantiated our results.

## 4. Discussion 

### 4.1. Summary of Main Findings

This systematic review and meta-analysis investigated the effects of DMI on psychological health (primary outcome) in older adults without dementia. In total, 14 RCT with 983 participants (*n-DMI* = 494, *n-control* = 489) were synthesized. Our analyses revealed a small positive effect of DMI on psychological health compared to control conditions. In addition, DMI had a medium effect on general cognitive function, assessed as an additional outcome, over comparators. The DMI varied in type, frequency and duration, resulting in high variability across studies. Our findings predominantly relate to clinically normal older adults and passive control conditions. Overall, DMI can be considered a promising multimodal enrichment tool to promote mental health and well-being in the older population. High-quality intervention studies are needed to expand evidence on DMI-induced changes in specific psychological domains and identify the underlying neurophysiological correlates.

### 4.2. DMI and Psychological Health as Primary Outcome 

We demonstrate that DMI have a small positive effect on overall psychological health (*g* = 0.30) compared to control conditions in older adults without dementia. Re-running analyses after the exclusion of outliers corroborated the effect (*g* = 0.37). The finding converges with the previous meta-analysis across different age groups by Koch and colleagues [44], showing an overall positive effect of DMI on combined psychological outcomes with a medium effect size. Further evaluating the positive psychological domain, we found a small effect at trend level in favor of DMI over comparators in well-being (*g* = 0.30), whereas the small effect of DMI on QoL was not statistically significant (*g* = 0.29). The exclusion of outliers substantiated both effects (*g* = 0.46, *g* = 0.37), respectively. In general, these results are in line with those of previous evidence syntheses. Wu and colleagues [46] have reported a significant positive effect of DMI compared to comparators on QoL in persons with MCI, which was, however, not found by others [47]. Syntheses of DMI studies across different age groups have further demonstrated improvements in subjective well-being, positive affect, QoL and interpersonal skills following DMI compared to control interventions [43,44]. 

Less clear results have been obtained for the negative psychological domain in previous primary studies [82,83]. In the present meta-analysis, the small effects of DMI on depression (*g* = 0.22) and anxiety/stress (*g* = 0.30) in older adults without dementia were statistically non-significant. The most promising evidence for DMI in this domain mainly relates to clinical populations, including patients with dementia, showing a reduction of depressive symptoms in response to DMI [84,85]. Evidence synthesis across different age groups showed medium effects of DMI with high heterogeneity in the reduction of depression and anxiety [44]. Interestingly, we observed a large positive effect of DMI on social integration/connectedness (*g* = 0.85). Despite limited reliability due to the small number of studies, this finding deserves further investigation since social isolation is suggested as a risk factor for AD [5]. Recent pilot trials have reported enhanced social connectedness after a single online DMI session in adults aged ≥ 18 years [86] and a reduction in feelings of social isolation in older adults with SCD/MCI after a 3-month-long movement program based on principles of creative arts therapy [87]. DMI might thus be expected to foster social integration and participation in older adults, which are shown to be protective factors against dementia [27].

### 4.3. DMI and Cognitive Function as Additional Outcome

Our meta-analysis further demonstrates a positive medium effect of DMI on cognitive function compared to control conditions (*g* = 0.50). In the present study, we analyzed general cognitive abilities as an additional outcome because this was not included in the search terms. Therefore, the observed effect is limited to the data reported in the primary trials, which reported psychological health outcomes and cognitive function. Since we did not conduct an independent search for cognitive outcomes, there may be a bias in the data selection procedure. The present result replicates prior meta-analytical studies, showing robust beneficial effects of DMI on several cognitive domains, including executive functions [88], memory [38] and global cognition [37] in clinically normal older adults and in participants diagnosed with MCI [39,89]. 

### 4.4. Putative Neurophysiological Mechanisms 

None of the primary studies has evaluated measures of physiological changes and neuroplasticity, thus limiting our understanding of the candidate mechanisms underlying the positive effects of DMI on psychological health outcomes. Comprehensive reviews have suggested that dance-based activities could promote neural plasticity in distributed brain regions and networks [35,90,91], some of which are involved in emotion regulation and processing. For example, interventional studies in older adults have shown increased functional connectivity in higher-order cognitive/emotional systems, including the default mode and fronto-parietal networks, in response to 3-, 4- and 6-month-long dance-based programs, respectively [87,92,93]. Other studies in similar populations have demonstrated modifications in white matter integrity in the fornix [94] and gray matter morphology in response to DMI compared to comparators. The latter encompassed distributed brain regions, including the cingulate cortex, insula and corpus callosum (after 6 months) along with the hippocampus (after 18 months) [95], although another study reported no change in hippocampal volume after a 4-month-long dance program [96]. Interestingly, one intervention study has found reduced salivary cortisol concentrations during the cortisol awakening response, as a marker of chronic stress, after a 3-month-long dance movement program compared to aerobic exercise [97]. Whether and how these multilevel physiological changes are related to the psychological benefits of DMI in older populations requires further investigation. 

### 4.5. Recommendations for Future Studies 

Our results support the view that DMI could act as a multimodal enrichment tool to promote psychological health and well-being in older adults without dementia. The reported adherence rates in the primary studies were high, suggesting that older participants who attended DMI programs were motivated and compliant. This fact may facilitate the transfer of lifestyle activity into everyday life, which in turn may help to sustain the health benefits of the intervention [34]. However, the DMI programs varied in type, frequency and duration, resulting in a high variability across studies. Due to the limited availability of primary studies and outcome measures, we could not perform a meta-regression analysis in the present synthesis. Previous authors have, however, suggested that longer intervention durations might be more effective in improving psychological outcomes, given that the exposure to the multimodal stimulation associated with DMI, including social interaction and connectedness, is prolonged [46]. Above this, the type of DMI could be an important determinant of psychological benefits. Evidence synthesis by Koch and colleagues [44] has shown a moderation of the type of DMI, with dance/movement therapy yielding better effects on psychological outcomes in a broader sample of studies. Overall, more research is required to draw reliable conclusions about the effectiveness of the different DMI features in older adults.

Our findings indicate the need for high-quality trials to evaluate the effects of DMI on psychological health measures as designated primary outcomes. Across the included studies, we observed high statistical heterogeneity and risk of bias across studies, as documented previously [44]. Whilst a relatively high proportion of studies were rated at a high overall risk of bias, this can be attributed to the assessment algorithm of the RoB 2 tool, which downgrades studies with self-reported outcomes. Another explanation relates to the nature of non-pharmacological interventions, where blinding is only possible to a certain extent. To distinguish between nonspecific and specific intervention effects, active control groups (receiving an effective alternate treatment) are needed. Furthermore, we found that data regarding the long-term effects of DMI were scarce, indicating the necessity of follow-up measurements. We also noticed that the vast majority of participants were female. This should be considered in trial protocols by tailoring recruitment strategies to attract more male participants. Finally, as mentioned above, the neurophysiological correlates underlying DMI-induced changes in psychological health need to be investigated. This will foster a bidirectional view of the body (including the brain) and mind as integrated and interacting parts of an adaptive system, which should form the basis for future studies.

### 4.6. Synopsis and Outlook

It has been recognized that dance activities could serve as an embodied prevention strategy to promote reserve and resilience against dementia, including AD [35]. As an integrated mind–body activity, DMI encourages the activation and synchronization of multimodal motor, sensory, cognitive and social processes, which is proposed to act as a protective factor against age- and disease-related conditions [35]. The positive impact of DMI on psychological health and well-being, as highlighted by the present study, could be particularly important in the prevention or early intervention amongst older populations at increased risk of developing AD. Those include older adults diagnosed with SCD, a population characterized by elevated negative psychological burden and/or altered functioning in brain networks subserving emotion regulation and self-referential thought [13,98,99]. Yet, none of the synthesized studies has specifically investigated older participants with SCD as a designated target group for early non-pharmacological intervention strategies to enhance resilience and/or reduce incipient symptoms [100,101]. 

Based on the current evidence, we perceive a window of opportunity to develop and evaluate a new generation of DMI for older people. As guiding principles, such intervention programs should 1. specifically aim to promote well-being on multiple levels, including physical, mental/psychological, and social well-being; 2. be easily accessible by their implementation into onsite and online settings [86]; and 3. be tailored to the needs of clinical and non-clinical older populations [71,87]. For example, incorporating dance/movement therapy, mindful movement practices and psychological techniques into targeted DMI programs may enhance mental skills, such as creativity, positive thinking and mindfulness, which have far-reaching benefits for the brain, cognitive and mental health in older adults [25,26,102,103]. Mastering new skills may further improve self-efficacy in older people, which in turn could strengthen feelings of mastery and autonomy as essential factors in coping with age-related conditions and adverse events [10,104,105]. Together such strategies could pave the road towards more effective and accessible multimodal lifestyle-based intervention strategies with high adherence to promote healthy mental aging and prevent AD in the long term. 

### 4.7. Strengths and Limitations

To our knowledge, this is the first systematic review and meta-analysis investigating the effect of DMI on psychological health outcomes in older adults without dementia. The review was strictly conducted according to a pre-specified open-source protocol, as stipulated by relevant guidelines [48]. Screening, risk of bias assessment and data extraction were performed in parallel by two independent reviewers to ensure the accuracy of the data. Bias rating was conducted with the revised and updated Risk of Bias 2 tool, which was designed to assess bias more accurately in light of developments in the field [54]. Statistical analyses and data synthesis followed the approved methodology applied in the field [57]. Additionally, multiple effect sizes of studies were considered in our meta-analytic approach with robust variance estimation.

One of the major limitations of the present synthesis is the small number of included studies. The search yielded a few high-quality RCT in older adults that have included psychological health outcomes. Numerous studies had to be discarded due to methodological limitations or insufficient data documentation. The sample sizes of primary studies were rather small, even though this was accounted for with a bias correction factor when pooling the data. One might argue that the number of databases searched was limited. However, the vast majority of eligible studies are covered by the three databases (PubMed, Web of Science and PsycINFO) as they encompass the fields of medicine, science, psychology and psychological aspects of many disciplines. In addition, the databases were reviewed with regard to academic publications (dissertations) to increase the scope of the search. Furthermore, we synthesized studies of older populations without a clinical diagnosis of depression or anxiety. Effects of DMI on psychological health may be more pronounced in participants with clinically relevant psychological symptoms [82,106,107]. Another limitation relates to the rather wide age range (55+) of participants who were considered to be eligible, which might have concealed intervention effects distinct to older age groups [37]. However, this was necessary to obtain a sufficient number of trials to be included in the present meta-analysis. Finally, due to the broad definition of DMI, the intervention programs differed in style and content. Nevertheless, all types of DMI share common principles with an emphasis on specific key features, which justifies the synthesis of data.

## 5. Conclusions

This systematic evaluation of 14 primary RCT studies suggests that, compared to comparators, DMI improves overall psychological health in older adults without dementia. Thus, DMI may serve as a promising tool in the promotion of healthy aging and early intervention of age-related conditions. The implementation of high methodological standards and well-designed assessments of psychological health outcomes and neurophysiological mechanisms of action should be a key goal in future trials. Effective transfer of this lifestyle activity into everyday life could be expected to help retain the positive effects of DMI on mental health outcomes, which needs investigation in studies of longer duration.

## Figures and Tables

**Figure 1 brainsci-13-00981-f001:**
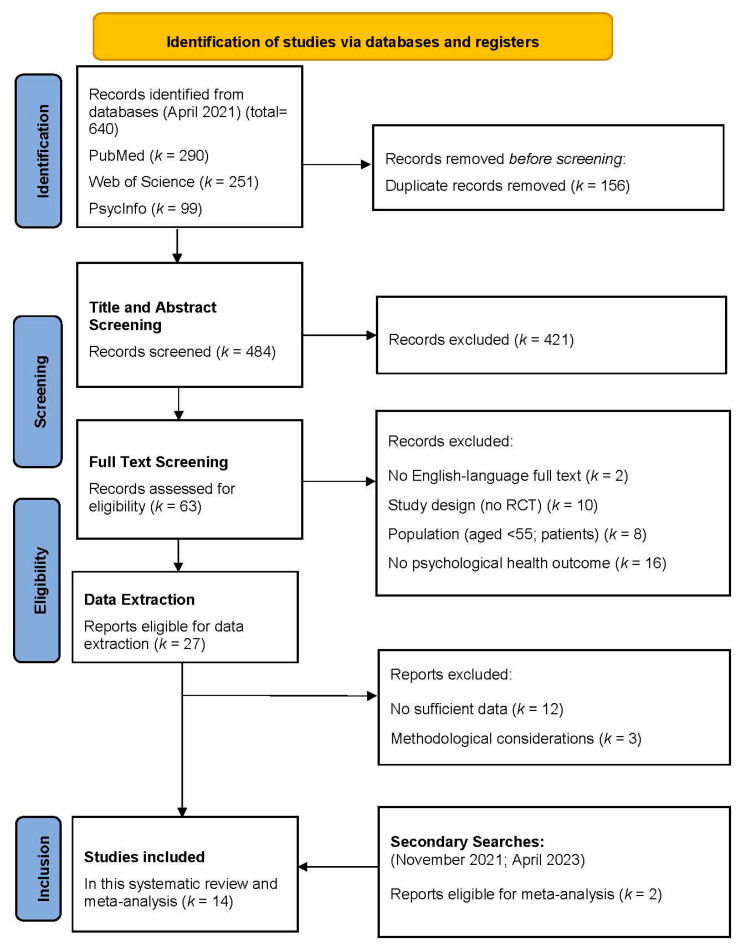
PRISMA flowchart. **Key**. *k*, Number of studies; *RCT*, randomized control trial.

**Figure 2 brainsci-13-00981-f002:**
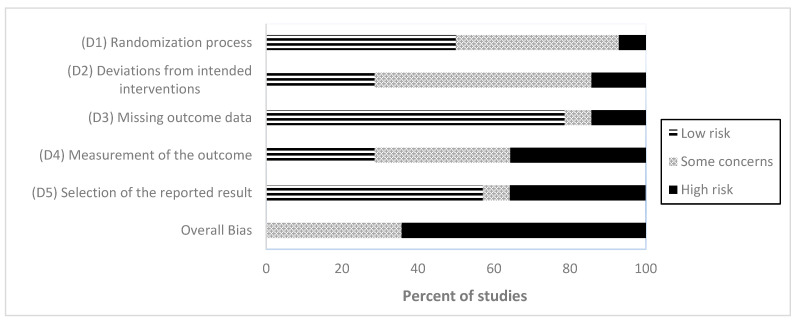
Summary of Risk of Bias Rating according to domains. Note. The risk of bias graph presents ratings for the 14 included studies rated regarding an intention-to-treat approach.

**Table 2 brainsci-13-00981-t002:** Meta-analyses comparing DMI to passive control conditions (where applicable) for psychological health outcomes, Quality of Life and cognitive function.

	Construct	k *(N* ES)	ES	95% *CI*	SE	df	*p*-Value	*I*^2^%
Overall psychological health	All combined	14 (33)	0.30	[0.06, 0.53]	0.11	12	0.02	65.04
Positive domain	Well-being	9 (16)	0.30	[−0.03, 0.63]	0.14	8	0.07	62.80
	Social Integration	3 (3)	0.85	[−1.59, 3.29]	0.57	2 *	*	85.17
Negative domain	Anxiety/Stress	4 (5)	0.30	[−0.54, 1.14]	0.26	3 *	*	66.22
	Depression	5 (5)	0.22	[−0.25, 0.68]	0.16	4	0.26	60.81
Quality of Life	All combined	9 (22)	0.29	[−0.14, 0.73]	0.19	8	0.16	76.10
General cognitive function	All combined	8 (9)	0.50	[0.12, 0.89]	0.16	7	0.02	79.61

Key: DMI = Dance Movement Interventions; k, number of studies; *N* ES, Number of Effect Sizes; ES, Effect size; CI, Confidence interval; SE, Standard Error; df, Degrees of freedom. * Where df < 4, *p*-values are unreliable and are thus not reported.

## Data Availability

The data from the primary studies summarized in this meta-analysis are provided in the article/Appendix A. Further requests can be directed to the corresponding author (OP). The R scripts necessary to reproduce the present analyses are available from the corresponding author (OP).

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
