# Peer review of "The Impact of Dance Movement Interventions on Psychological Health in Older Adults without Dementia: A Systematic Review and Meta-Analysis"

_brainsci, 2023, doi:10.3390/brainsci13070981_

Round 1
Reviewer 1 Report
Comments and Suggestions for Authors
I had the opportunity to review this very interesting systematic review evaluating the effects of DMIs on the mental health of older adults. The study is relevant, and the findings have practical implications. Also, the study is methodologically sound, and the manuscript is well-written. Nonetheless, I have four major comments on the study needing further clarification.
First, usually, people who are +60 years are considered old. So, the rationale for considering those who are +55 as old needs further clarification.
Second, it is more than one year since the authors last conducted their search (November 2021). As new relevant articles might have been published during this time, I suggest performing the search again and updating the results accordingly.
Third, some crucial databases, such as Embase and CENTRAL, were not searched, which is a major limitation of this study.
Finally, there are issues with the meta-analysis in Figure 3. The analysis considers a variety of measures as proxies of psychological health. For example, quality of life has been included in the analysis, but the quality of life is not a measure of psychological health, despite being associated with it. I would suggest omitting this figure and focusing on outcome domains individually.
Reviewer 2 Report
Comments and Suggestions for Authors
I have carefully reviewed the manuscript, titled “The impact of dance movement interventions on psychological health in older adults without dementia: A systematic review and meta-analysis”. The study was to examine the efficacy of dance movement interventions (DMI) on a basis of meta-analysis, as an integrated mind-body activity, on outcomes of psychological health in older adults.
The study has been substantially revised and some strong points (clear introduction, accurate statistical analysis, constructive discussion).
However, I would like to ask the authors to address some points in order to improve the paper
Introduction:
1) Can you present some psychological explanations for psychological health? Why do people experience it in general? What are its underlying mechanisms (p. 2)
2) As the study examines the impact of dance movement interventions on psychological health and well-being, it would be appropriate to provide more information on the role of well-being in older adults, which is included in the following works:
– https://doi.org/10.1080/10508619.2018.1556061 (When meaning matters: Coping mediates the relationship of religiosity and illness appraisal with well-being in older cancer patients)
– doi:10.1017/S1041610220000964 (Psychological well-being among older adults during the COVID-19 outbreak: A comparative study of the young–old and the old–old adults)
Method:
3) How were the types of interventions chosen? (p. 4)
4) How did you handle missing values in your data? (If any exist)
Results:
5) The results are properly showed.
Discussion:
6) What are the underlying mechanisms responsible for this result: “Longer duration might foster effectiveness of the intervention on psychological health outcomes” (p. 16)?
7) Can you elaborate on the following statement in relation to the bias mentioned: “Therefore, the effect is limited to the data reported in the primary trials, as we did not conduct an independent search for this main outcome. Hence, there may be a bias in the data selection (p. 17)”. Please, provide a potential explanation of this bias as it is an interesting result. Is it a matter of cognitive bias or something else?
Round 2
Reviewer 1 Report
Comments and Suggestions for Authors
I would like to thank the authors for revising the manuscript according to my comment. The manuscript's quality has improved after the revisions, and I have no further comments.